# Contrasting Litter Nutrient and Metal Inputs and Soil Chemistry among Five Common Eastern North American Tree Species

Neil F. J. Ott [1],* and Shaun A. Watmough [2]

1   Environmental and Life Sciences Graduate Program, Trent University, Peterborough, ON K9J 7B8, Canada
2   School of the Environment, Trent University, Peterborough, ON K9J 7B8, Canada; swatmough@trentu.ca
*   Correspondence: neilott@trentu.ca

**Abstract:** Forest composition has been altered throughout Eastern North America, and changes in species dominance may alter nutrient cycling patterns, influencing nutrient availability and distribution in soils. To assess whether nutrients and metals in litterfall and soil differed among sites influenced by five common Ontario tree species (balsam fir (*Abies balsamea* (L.) Mill.), eastern hemlock (*Tsuga canadensis* (L.) Carr.), white pine (*Pinus strobus* L.), sugar maple (*Acer saccharum* Marsh.), and yellow birch (*Betula alleghaniensis* Britt.)), litterfall and soil chemistry were measured at a managed forest in Central Ontario, Canada. Carbon (C) and macronutrient (nitrogen (N), phosphorus (P), potassium (K), calcium (Ca), and magnesium (Mg)) inputs in litterfall varied significantly among sites, primarily due to differences in litterfall mass, which was greatest in deciduous-dominated sites, while differences in elemental concentrations played relatively minor roles. Trace metal inputs in litterfall also varied, with much higher zinc (Zn) and cadmium (Cd) in litterfall within yellow birch dominated stands. Mineral soil oxide composition was very similar among sites, suggesting that differences in soil chemistry were influenced by forest composition rather than parent material. Litter in deciduous-dominated stands had lower C/N, and soils were less acidic than conifer-dominated sites. Deciduous stands also had much shorter elemental residence times in the organic horizons, especially for base cations (Ca, Mg, K) compared with conifer-dominated sites, although total soil nutrient pools were relatively consistent among sites. A change from stands with greater conifer abundance to mixed hardwoods has likely led to more rapid cycling of elements in forests, particularly for base cations. These differences are apparent at small scales (100 m$^2$) in mixed forests that characterize many forested regions in Eastern North America and elsewhere.

**Keywords:** litterfall; forest composition; nutrients; trace metals; human disturbance



## 1. Introduction

Globally, humans have impacted the natural environment through changes in land use and forest management, altering forest composition, structure, and soil chemistry [1–3]. In Central Ontario, selective harvesting of red oak (*Quercus rubra* L.), eastern hemlock (*Tsuga canadensis* (L.) Carr.), and white pine (*Pinus strobus* L.) created ideal conditions for shade tolerant species, such as sugar maple (*Acer saccharum* Marsh.), to proliferate [4]. These practices have led to an increase in mixed–hardwood forests across much of this region and beyond, at the expense of the hemlock and pine dominated stands that were previously abundant [4–6]. Forest composition is continuously changing, and management of mixed hardwood forests generally favors the regeneration of multiple species including sugar maple, yellow birch (*Betula alleghaniensis* Britt.), balsam fir (*Abies balsamea* (L.) Mill), and American beech (*Fagus grandifolia* Ehrh.) [7].

Tree health can be influenced by the availability of essential macronutrients [8] or excess of micronutrients (e.g., manganese (Mn)) [9] and non-essential elements (e.g., aluminum (Al)) [10]. While management practices can influence the species that are intended

to regenerate in an area, nutrient availability also regulates tree growth [11] (pp. 92–118). Litterfall acts as an important pathway of nutrient and trace metal return to the forest floor [12,13], particularly for calcium (Ca) which is a key structural component of leaf tissue [14], and to a lesser degree for magnesium (Mg) [12,15]. Other elements such as nitrogen (N), phosphorus (P), and potassium (K) are translocated into perennial tissues prior to leaf fall so maximum foliage elemental concentrations are reached earlier in the growing season [15]. The extent to which trace metals are cycled through litterfall also differs from the macronutrients. Metals such as cadmium (Cd), lead (Pb), and zinc (Zn) are rapidly cycled through vegetation and accumulate in soils, where they can be toxic to terrestrial organisms [16].

Tree species vary widely in their nutrient requirements, and elemental cycling patterns can differ substantially among species [17,18]. Coniferous species have a greater ability to acidify soils than deciduous species, leading to increased metal solubility in upper soil horizons [19]. For example, eastern hemlock can create low–nutrient, acidic environments [20] and was found to be more abundant than sugar maple on soils with lower Ca and Mg [21]. On the other hand, sugar maple is a Ca demanding species [21] that can enrich the soil Ca content through litterfall [17]. In New England hardwood forests, foliar concentrations of essential elements (Ca, Mg, K, Mn, Zn) in deciduous species were up to 60% greater than in coniferous species, while coniferous stands had 30–50% smaller organic horizon pools of these elements than deciduous species [22]. Thus, these differing nutritional requirements suggest that elemental cycling through litterfall may differ considerably among species.

Prior to European settlement, white pine and eastern hemlock were more abundant in forests across Central Ontario, and these species grew on acidic, sandy soils of moderate to low fertility [23,24]. Shifts to mixed hardwood forests with an increasing proportion of deciduous species may increase nutrient availability, allowing for more rapid tree growth and increased carbon (C) sequestration, which has been predicted for other deciduous species [25]. However, this change could also have other ecosystem impacts, such as the decline of species that rely on conifers for survival [4]. It is necessary to identify the litterfall chemistry of different tree species to assess how changes in forest composition may influence soil chemistry over time. Elemental cycling may have been influenced by the proliferation of shade–tolerant hardwoods like sugar maple across Eastern North America [17,18,26]. Fulfilling this knowledge can help forest managers understand how future changes to forest composition may influence nutrient and metal availability in soils [27].

This study compared litterfall nutrient and metal inputs along with soil chemistry among five common tree species in Central Ontario, Canada. Research was conducted at the Haliburton Forest and Wild Life Reserve, where forest composition consists of numerous species including sugar maple, eastern hemlock, and yellow birch, with smaller stands of balsam fir and white pine. These five species were selected for this research. Traps were set out to collect litterfall, and soil samples were taken from sites dominated (but not pure stands) by each of these five tree species. It was expected that sites dominated by deciduous trees (sugar maple and yellow birch) would have greater nutrient inputs in litterfall compared with coniferous species, which have lower nutrient demands (e.g., white pine, eastern hemlock, and balsam fir). As a result, it was expected that macronutrients in both the organic and mineral soils would be notably greater at sites dominated by deciduous trees as opposed to conifer-dominated stands. It was also expected that soils in conifer-dominated stands would accumulate more trace metals and be more acidic than deciduous stands.

## 2. Materials and Methods

### 2.1. Study Area

Field work was conducted at the Haliburton Forest and Wild Life Reserve ($45°13'21''$ N, $78°35'32''$ W; now referred to as Haliburton Forest) in Haliburton County, approximately 200 km northeast of Toronto, Ontario. The forest, located in the Great Lakes–St. Lawrence

Forest region, is predominately temperate hardwood composed of sugar maple (*Acer saccharum* Marsh.), American beech (*Fagus grandifolia* Ehrh.), eastern hemlock (*Tsuga canadensis* (L.) Carr.), and yellow birch (*Betula alleghaniensis* Britt.), often in mixed stands. Annual precipitation for the Haliburton region is 1074 mm while the annual temperature is 5.0 °C (Station ID: 6163171) [28]. Soils are generally acidic (pH $\leq$ 4.7 in the Ah horizon) and belong to the Orthic or Eluviated dystric brunisol subgroups [29]. Overburden varies from thin till (<0.3 m in many places) to minor till (~1 m) on Precambrian Shield granitized hornblende and biotite gneiss, with outcrops present across the landscape [30].

### 2.2. Plot Construction and Field Methods

Stands dominated by balsam fir, eastern hemlock, white pine, sugar maple, or yellow birch were located throughout the Haliburton Forest. These stands are referred to as fir, hemlock, pine, maple, and birch stands, and they are not always pure stands, but are dominated by a single species (Table 1; additional plot characteristics shown in Table S1). Plots (10 × 10 m) were chosen so that the species of interest composed at least 75% of canopy cover, which resulted in the species composing >50% of the total plot basal area. At least three larger trees of that species (>25 cm diameter at breast height (dbh)) were included within each plot. At the beginning of September 2018, five plots were established for maple and pine, while three plots were established for fir, pine, and birch. Plots were located within close proximity to each other (<30 km) and were established on relatively flat ground (slope < 5°). Plots did not appear to have been recently harvested based on observations of tree size, density, and the absence of disturbance by machinery.

**Table 1.** Species composition of each plot in percent of total basal area of trees with a dbh >10 cm.

| Plot Number | Balsam Fir | Eastern Hemlock | White Pine | Sugar Maple | Yellow Birch |
| --- | --- | --- | --- | --- | --- |
| 1 | BF 79%, SM 21% | EH 76% WC 23% BF 1% | WP 95% AB 4% EH 2% | SM 65% BF 35% | YB 95% BF 5% |
| 2 | BF 58% SM 24% YB 19% | EH 100% | WP 74% WB 12% Other 11% BF 3% | SM 81% WB 15% AE 4% | YB 93% WB 7% |
| 3 | BF 53% SM 21% BA 26% | EH 100% | WP 83% BF 17% | SM 93% YB 5% BF 2% | YB 87% BF 13% |
| 4 | – | – | WP 100% | SM 92% BC 5% YB 1% WB 1% BF 1% | – |
| 5 | – | – | WP 100% | SM 100% | – |

Species abbreviations: AB = American Beech, AE = American Elm, BC = Black Cherry, BF = Balsam Fir, BA = Black Ash, EH = Eastern Hemlock, SM = Sugar Maple, WB = White Birch, WC = White Cedar, WP = White Pine, YB = Yellow Birch.

Basal area was determined by measuring the diameter of each tree greater than 10 cm dbh within the plot. Fir stands had the greatest basal area, where numerous younger balsam fir trees (dbh between 10–15 cm) were often clustered together within the understory. Within each plot, three rectangular litterfall collectors were placed on level ground to ensure accurate litterfall inputs per unit area were calculated. Litterfall was collected over the course of the Autumn (September–November), when the majority of annual litterfall reaches the forest floor (e.g., ~80% in September and October) [12,31]. In November, once the trees were bare, litterfall was collected from the plots.

At each plot corner, soils were sampled by horizon (four soil samples per horizon per plot). Litter (L) and fibric–humic (FH) horizons [29] were collected and placed into separate bags. Mineral soils were sampled from the Ah and upper and lower portion of the Bm horizon (Ah, $Bm_1$, and $Bm_2$) [29], and horizon depths were recorded. Samples

from the C horizon were also taken for analysis of soil oxides, to identify parent material characteristics among the stands. At the plot center, bulk density samples were obtained. A square frame was placed on the forest floor and the depth of each organic horizon was recorded. Mineral soil samples were obtained using a bulk density hammer, and samples were refrigerated (4 °C) for less than one week prior to analyses.

### 2.3. Litter and Soil Analyses

Leaf litter was oven dried (60 °C for 24 h) and sorted by coniferous needles, deciduous leaves, and woody debris (seeds, twigs, cones). Dry samples were weighed, and leaf litter was combined from the three litterfall traps in each plot, and bulked by coniferous, deciduous, and woody debris. Litter was then coarsely ground in a Wiley Mill (Model ED–5) and a further ~5 g subsample was pulverized for analyses using a coffee grinder. Organic and mineral soil samples were oven dried (105 °C for 24 h), ground, and sieved to remove particles greater than 2 mm in diameter. Soil samples collected from the four plot corners were bulked into one composite sample for each horizon per plot. Soil pH was determined at a 1:5 soil to reverse osmosis water slurry ratio. Samples were shaken for two hours and left to sit for an additional hour, before pH was measured with an Oakton pH MVC meter. Soil organic matter content was determined through the loss on ignition method. A 5 g sample of mineral soil (1 g for organic horizons) was heated in a muffle furnace at 450 °C for eight hours.

Pulverized leaf litter was packed into foil pellets and combined with tungsten (1:2 leaf litter: tungsten ratio) for C and N analysis using an Elementar vario MAX Cube (Elementar, Langenselbold, Germany). Tungsten was also added to the soil samples, except those from the Bm horizon. Elemental concentrations in leaf litter were determined through acid digestion. A 0.2 g sample of pulverized organic/mineral material was placed into a 50 mL DigiTUBE (SCP Science, Baie–D'Urfe, QC, Canada) and 2.5 mL of trace metal grade nitric acid ($HNO_3$, Aristar®Plus CAS 7697-37-2, VWR International, Radnor, PA, USA) were added. The tubes were swirled and placed on a digestion block for eight hours at 100 °C, with cold digestion proceeding for another eight hours following hot digestion. Samples were then rinsed with reverse osmosis water and passed through a funnel lined with Fisher P8 (Thermo Fisher Scientific, Waltham, MA, USA) filter paper into a 25 mL volumetric flask. Following filtration, the digested contents on the filter paper were rinsed with reverse osmosis water into the flask until a volume of 25 mL was reached. Samples were diluted 1:10, and analyses for Ca, Mg, K, Al, Mn, P, Zn, strontium (Sr), copper (Cu), and Cd were performed through ICP-OES (Perkin Elmer Optima 7000 DV ICP–OES; Waltham, MA, USA). Elemental recovery was confirmed by standard reference material including Apple Leaf (NIST 1515 SRM, Gaithersburg, MD, USA) and soil standards (EnviroMat SS–1/SS–2; SCP Science, Baie–D'Urfe, QC, Canada), along with blanks every 25 samples. Similarly, soil samples from the L, FH, and Ah horizons at each plot were also digested and analyzed for Ca, Mg, K, Al, Mn, P, Zn, Sr, Cu, and Cd. Aluminum, Mn, Sr, Zn, Cu, and Cd concentration and pools were only evaluated in organic horizons as concentrations of these metals were much lower in mineral horizons.

A 1 M ammonium chloride ($NH_4Cl$) solution was used to determine the exchangeable cations in organic and mineral horizons. Organic (1 g) and mineral soil samples (5 g) were weighed into 50 mL centrifuge tubes, and 25 mL of $NH_4Cl$ were added to each tube. Samples were placed on a shaker table for 16 h and left to sit for an additional hour, before they were vacuum filtered through Fisher P8 filter paper in into a flask. An additional 25 mL of $NH_4Cl$ were added to the centrifuge tube to ensure removal of all soil from the tube walls and was passed through the filter. The filtrate was poured into a new 50 mL centrifuge tube. Exchangeable cation samples were diluted 1:10 and acidified with 0.2 mL of trace metal grade $HNO_3$. Analyses for Ca, Mg, K, Al, and Mn were performed through ICP-OES (Perkin Elmer Optima 7000 DV ICP–OES; Waltham, MA, USA).

To assess parent material characteristics among the stands, soils collected from the lower B ($Bm_2$) and C horizons were analyzed for total elemental oxides. Five grams of

pulverized soil from three plots of each of the five stand types (30 samples) were sent to SGS Canada (Lakefield, ON, Canada) for total elemental oxide analysis using X–ray fluorescence spectrometry. Duplicates, blanks, and standard reference materials confirmed the method credibility.

### 2.4. Data Analysis

Litterfall elemental inputs were calculated by multiplying elemental concentrations by litterfall mass, dividing by litterfall trap area ($0.13 \text{ m}^2$), and converting to input per unit area ($\text{mg m}^{-2}$, $\text{kg ha}^{-1}$, or $\text{t ha}^{-1}$). Differences in litterfall mass, analyte concentrations and total inputs, as well as individual inputs from coniferous, deciduous, and woody debris were investigated. In maple and birch stands, quantities of coniferous needle litter were insufficient to conduct analytical testing for elements, whereas enough deciduous leaf litter was collected in all plots for analysis. As well, insufficient quantities of woody debris were collected for analysis in pine stands.

Soil elemental pools were calculated by multiplying bulk density by horizon depth and elemental concentration. Soil properties including bulk density, soil mass per unit area, pH, organic matter content, and analyte concentrations/inputs were compared among stands for the L, FH, Ah, and Bm horizons ($Bm_1$ and $Bm_2$) separately. Differences in total soil elemental pools for each analyte were also examined, and soil oxide composition was statistically tested for the combined lower B ($Bm_2$) and C horizons.

Organic horizon elemental residence times (years) were calculated by dividing the total elemental mass in the organic horizon pool by the annual inputs in litterfall [32]. Residence times indicate the rate at which elements are cycled through the organic horizons in soil. Longer residence times are associated with a slower decomposition of elements and vice versa. Relationships between elemental inputs in litterfall and elemental pools in soil organic horizons were also assessed by calculating Pearson correlation coefficients. Elemental residence times were compared among stands, along with C/N ratios in the organic horizons. Pearson correlation coefficients were used to investigate the relationship between litterfall mass and stand basal area.

Differences among the five stand types in these litterfall and soil properties described were tested following assessment for normality using Shapiro–Wilk tests. Normally distributed samples were compared using one–way ANOVA tests, followed by Tukey's post hoc tests where applicable. Data transformations were made when normality was not met by logarithmically transforming the data. When these transformations did not result in normalized distributions, non-parametric Kruskal–Wallis tests were performed, followed by Dunn's post hoc tests where applicable. Statistical analyses were performed using the Real Statistics Resource Pack (Release 7.3.3) add-in to Microsoft Excel [33].

## 3. Results

### 3.1. Litterfall Characteristics

#### 3.1.1. Litterfall Mass

Litterfall mass differed significantly among sites, with maple contributing the greatest inputs ($3.2 \text{ t ha}^{-1}$) and pine contributing the lowest ($1.2 \text{ t ha}^{-1}$) (Figure 1; statistical test results shown in Table S2). Litterfall mass was not correlated with basal area ($R^2 = 0.01$; data not shown) ruling out the possibility greater litterfall mass was a result of greater stand basal area. As expected, deciduous leaf litter accounted for the majority of total litterfall mass in maple and birch stands (up to 95%). However, in conifer-dominated stands, needle litter played somewhat of a smaller role. Deciduous leaf fall was considerable in all stands (Figure 1), associated with the dominance of hardwoods in this region. Inputs of woody debris in litterfall were greatest in fir stands (17% of all litterfall) but accounted for less than 10% of litter mass in all other stands.

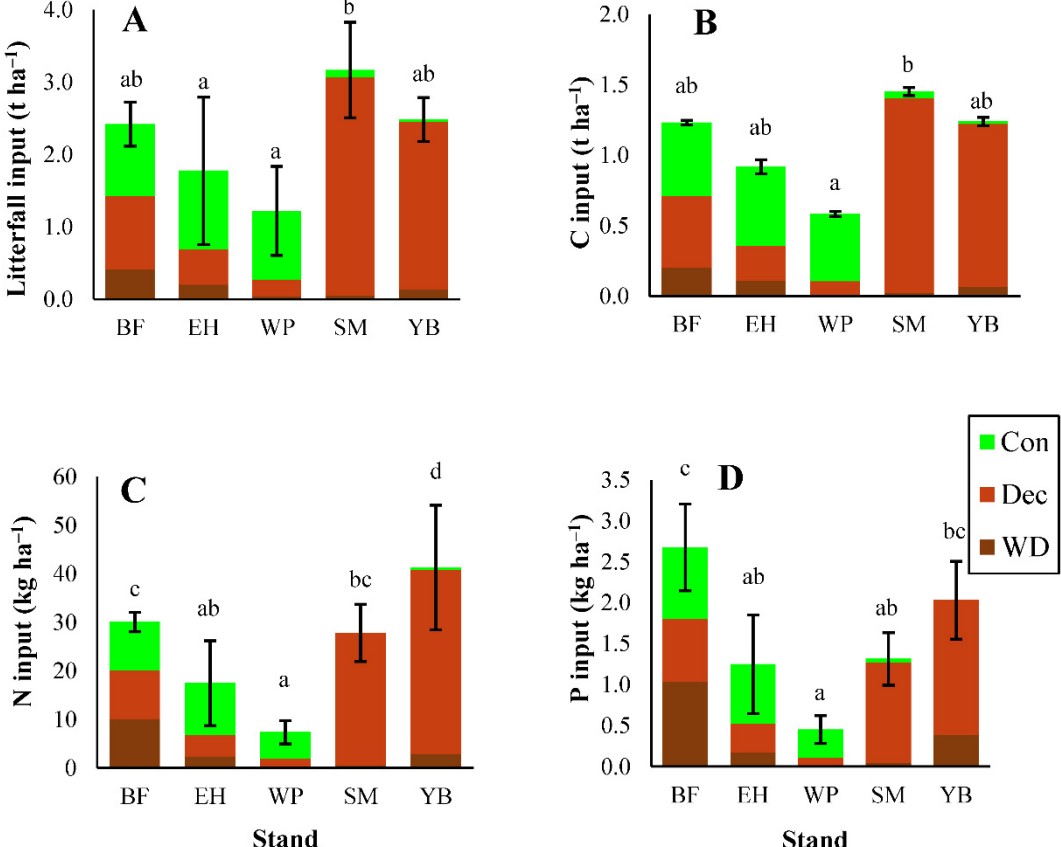

**Figure 1.** Litterfall mass (**A**) and inputs of carbon (C) (**B**), nitrogen (N) (**C**), and phosphorus (P) (**D**) in coniferous needles (Con), deciduous leaf litter (Dec), and woody debris (WD). BF = Balsam Fir, EH = Eastern Hemlock, WP = White Pine, SM = Sugar Maple, YB = Yellow Birch. Different letters represent significant differences ($p < 0.05$) in total litterfall inputs among stands. ANOVA test results shown in Tables S2 and S3. Error bars indicate standard error (S. E.). For BF, EH, and YB, $n = 3$. For SM and WP, $n = 5$.

### 3.1.2. Carbon and Macronutrients

Litterfall inputs of C closely matched patterns in litterfall mass across stands (Figure 1). Carbon inputs were greatest in maple stands (1.4 t C ha$^{-1}$) and lowest in pine stands (0.6 t C ha$^{-1}$; Figure 1; statistical test results shown in Table S3). The proportion of C in litterfall ranged from 44% in maple stands to 53% in hemlock stands (concentrations shown in Table S4). Litterfall inputs of N were the highest of all the macronutrients and ranged from 7–41 kg N ha$^{-1}$ among stands (Figure 1; concentrations in Table S5). Deciduous leaf litter in maple and birch stands contributed the greatest quantity of N to the forest floor (27 kg N ha$^{-1}$ and 38 kg N ha$^{-1}$). In fir stands, woody debris delivered about one third of total N litterfall inputs to the forest floor (10 kg N ha$^{-1}$). Phosphorus inputs in litter were primarily influenced by litterfall mass rather than concentrations (Figure 1, Table S6). Fir stands had the greatest P inputs (2.7 kg P ha$^{-1}$) while pine had the lowest (0.5 kg P ha$^{-1}$; Figure 1). Phosphorus concentrations were generally higher in woody debris compared with leaf litter (Table S6), and fir stands had the greatest inputs of woody debris (Figure 1).

Base cation (Ca, Mg, K) inputs in litterfall followed the order of Ca > K > Mg, in all stands except for birch, where Mg > K (Figure 2). Calcium inputs in litterfall followed patterns in litterfall mass and were higher in the deciduous stands than conifer stands. Calcium inputs were also notably lower in pine stands (~8 kg Ca ha$^{-1}$; Figure 2), primarily associated with low litterfall mass. Overall, base cation concentrations were substantially lower in all litter types from pine stands compared with the other stands (Tables S7–S9). Magnesium and K inputs in litterfall generally followed a similar pattern to Ca, with larger inputs in deciduous stands and low inputs in pine stands (Figure 2). However, Mg inputs

were elevated in birch stands due to higher Mg concentrations in deciduous leaf litter and woody debris (Table S8). In all stands, deciduous leaf litter transferred a large component of base cations from the canopy to the forest floor (Figure 2).

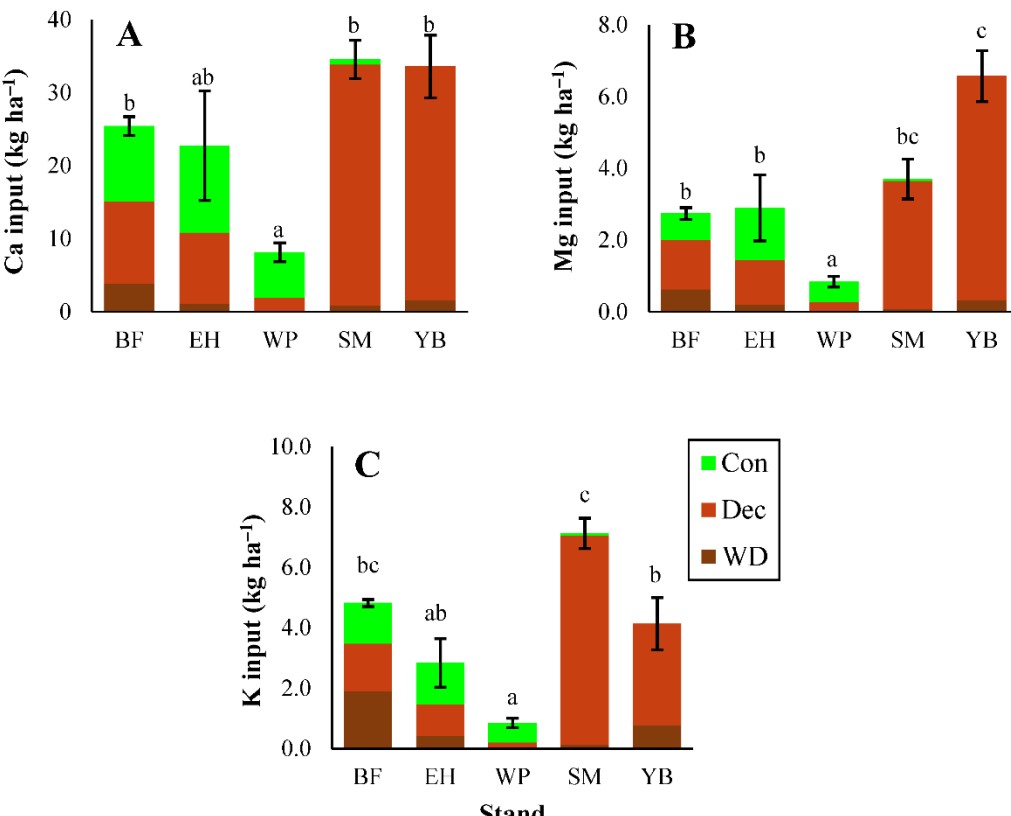

**Figure 2.** Inputs of calcium (Ca) (**A**), magnesium (Mg) (**B**), and potassium (K) (**C**) in each litterfall component (Con, Dec, WD). Different letters represent significant differences ($p < 0.05$) in total litterfall inputs among stands. ANOVA test results shown in Table S3. Error bars indicate standard error (S. E.). For BF, EH, and YB, $n = 3$. For SM and WP, $n = 5$.

### 3.1.3. Micronutrients and Trace Metals

Trace metal inputs through litterfall generally followed the order of Mn > Al > Zn > Sr > Cu > Cd with some exceptions (Figure 3). Litterfall inputs of Al were greater in coniferous stands than deciduous stands, but differences were only significant between hemlock and maple/birch stands (Figure 3; statistical test results shown in Table S10). Coniferous needle litter was responsible for the majority of Al inputs in conifer-dominated stands. Manganese inputs in litterfall did not differ significantly among stands (Figure 3), but among the three litter types, deciduous leaf litter provided the greatest Mn inputs to the forest floor in all stands except pine (Figure 3). Birch stands had elevated litterfall inputs of Zn and Cd due to the higher concentrations in deciduous leaf litter (micronutrient and trace metal concentrations shown in Tables S11–S16). For all trace elements except Al, inputs in litterfall were lowest in pine stands, but these differences were only significant for Sr and Cu (Figure 3). Overall, total trace metal inputs in litterfall tended to be greater in deciduous-dominated stands, except for Al where inputs were greatest in conifer stands.

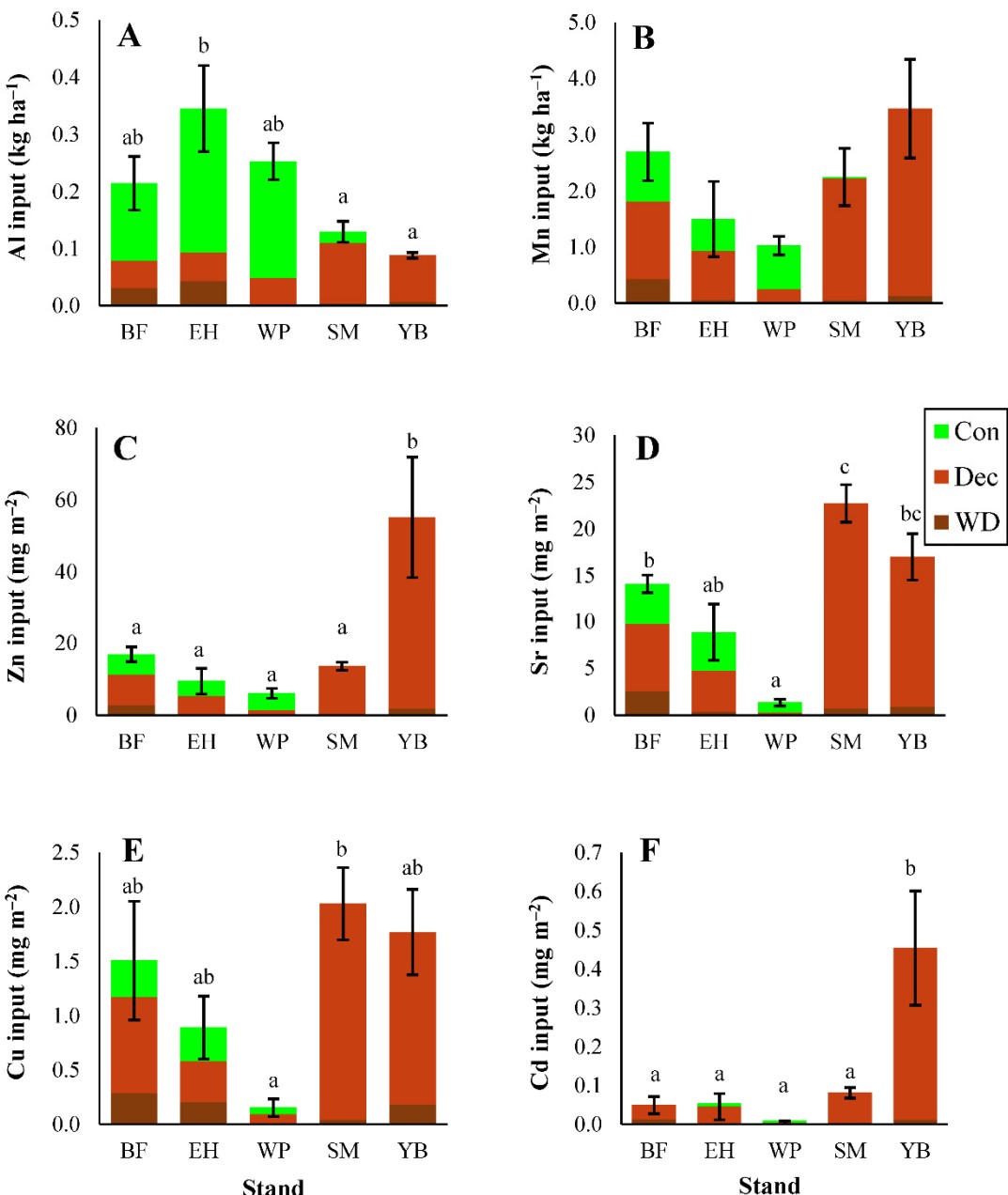

**Figure 3.** Inputs of aluminum (Al) (**A**), manganese (Mn) (**B**), zinc (Zn) (**C**), strontium (Sr) (**D**), copper (Cu) (**E**), and cadmium (Cd) (**F**) in each litterfall component (Con, Dec, WD). Different letters represent significant differences ($p < 0.05$) in total litterfall inputs among stands; where letters are not present, significant differences do not exist. ANOVA test results shown in Table S10. Error bars indicate S. E. For BF, EH, and YB, $n = 3$. For SM and WP, $n = 5$.

### 3.2. Soil Properties

3.2.1. Physical and Chemical Properties

Soil bulk density and horizon mass were generally similar among the stands, but hemlock stands had the greatest soil mass in the FH and Ah horizons (Table 2). All sites were acidic, and coniferous stands were more acidic than deciduous stands. Soil pH was greatest in the L–horizon and lowest in the Ah horizon. Soil pH differed significantly among stand types and was highest in maple stands for all soil horizons (range = 4.7–5.0) and lowest in hemlock and pine stands (range = 3.7–4.5). Soil organic matter content in the organic horizons was greatest in pine and hemlock stands but did not differ among stands in lower mineral horizons. In all stands, soil organic matter decreased with soil depth, averaging around 90% in the L horizon compared with 5–10% in the Bm–horizon.

**Table 2.** Soil properties within the five stand types (mean $\pm$ S. E.). Different letters represent significant differences among stands. For BF, EH, and YB, $n$ = 3. For SM and WP, $n$ = 5.

| Horizon | Parameter | Balsam Fir | Eastern Hemlock | White Pine | Sugar Maple | Yellow Birch | *p* Value |
|---|---|---|---|---|---|---|---|
| L | Bulk Density (kg m$^{-3}$) | 31 (15) | 34 (10) | 18 (4) | 39 (14) | 12 (5) | 0.468 |
| | Mass (kg m$^{-2}$) | 0.2 (0.1) | 0.2 (0.1) | 0.5 (0.1) | 0.2 (0.1) | 0.1 (<0.1) | 0.163 |
| | pH | 4.8 [ab] (0.2) | 4.5 [a] (0.2) | 4.5 [a] (<0.1) | 5.0 [b] (0.1) | 5.2 [b] (0.3) | <0.001 |
| | %OM | 92.4 [ab] (0.3) | 95.2 [a] (0.1) | 94.4 [ab] (1.0) | 87.3 [b] (1.5) | 91.6 [ab] (0.4) | <0.05 |
| FH | Bulk Density (kg m$^{-3}$) | 116 (22) | 54 (8) | 41 (16) | 89 (18) | 89 (16) | 0.060 |
| | Mass (kg m$^{-2}$) | 7.6 [a] (1.9) | 8.4 [a] (1.9) | 1.8 [b] (0.7) | 1.8 [b] (0.4) | 3.2 [ab] (0.9) | <0.01 |
| | pH | 4.4 (<0.1) | 4.4 (0.1) | 4.3 (0.1) | 4.9 (0.1) | 4.3 (0.1) | 0.055 |
| | %OM | 82.8 (7.5) | 93.3 (1.2) | 82.5 (4.5) | 75.5 (2.6) | 76.1 (13.5) | 0.238 |
| Ah | Bulk Density (kg m$^{-3}$) | 813 [ab] (127) | 793 [ab] (50) | 434 [a] (104) | 524 [ab] (51) | 915 [b] (100) | <0.05 |
| | Mass (kg m$^{-2}$) | 52.3 [ab] (11.2) | 70.0 [a] (19.3) | 21.7 [b] (5.2) | 26.2 [b] (2.5) | 50.8 [ab] (8.2) | <0.01 |
| | pH | 4.1 [ab] (0.3) | 3.7 [a] (<0.1) | 3.9 [ab] (0.2) | 4.7 [b] (0.1) | 4.2 [ab] (0.1) | <0.05 |
| | %OM | 5.0 [a] (2.0) | 4.1 [a] (0.9) | 20.1 [b] (0.3) | 12.5 [ab] (3.0) | 5.9 [a] (1.0) | <0.01 |
| Bm$_1$ | Bulk Density (kg m$^{-3}$) | 976 (212) | 944 (29) | 699 (84) | 803 (49) | 1018 (114) | 0.309 |
| | Mass (kg m$^{-2}$) | 66.3 (16.6) | 81.2 (35.5) | 55.9 (7.5) | 64.3 (3.9) | 61.1 (5.6) | 0.761 |
| | pH | 4.6 [a] (0.1) | 4.1 [b] (<0.1) | 4.1 [b] (0.1) | 4.8 [a] (0.1) | 4.5 [ab] (<0.1) | <0.001 |
| | %OM | 6.3 (2.5) | 8.5 (2.4) | 11.9 (2.0) | 8.3 (3.0) | 7.5 (1.1) | 0.610 |
| Bm$_2$ | Bulk Density (kg m$^{-3}$) | 1157 (121) | 990 (64) | 830 (72) | 953 (227) | 890 (165) | 0.677 |
| | Mass (kg m$^{-2}$) | 309.0 (32.3) | 270.3 (17.4) | 266.8 (63.5) | 266.9 (40.3) | 232.5 (20.1) | 0.812 |
| | pH | 4.7 [ab] (0.1) | 4.2 [a] (<0.1) | 4.3 [ab] (0.1) | 4.8 [b] (0.1) | 4.7 [ab] (0.1) | <0.05 |
| | %OM | 5.6 (1.8) | 8.9 (1.0) | 7.8 (1.7) | 6.8 (1.4) | 5.2 (0.2) | 0.818 |

Soil oxide concentrations in the Bm and C horizons were similar in each of the five stands and soils were dominated by silica ($SiO_2$), which composed 60–67% of total oxides (Table 3). There were no significant differences in base cation oxide proportions (calcium oxide (CaO), magnesium oxide (MgO), and potassium oxide ($K_2O$)), and differences were only indicated for iron(III) oxide ($Fe_2O_3$) and manganese(II) oxide (MnO) which were highest in maple and pine stands.

**Table 3.** Total oxides present in mineral soils (Bm and C horizons) within the five stand types (mean $\pm$ S.E.). Different letters represent significant differences among stands. For all stands, $n$ = 3.

| Oxide | Balsam Fir | Eastern Hemlock | White Pine | Sugar Maple | Yellow Birch | *p* Value |
|---|---|---|---|---|---|---|
| | | | Units = % of soil | | | |
| $SiO_2$ | 66.8 (1.8) | 64.7 (1.7) | 62.4 (3.5) | 60.7 (4.6) | 59.7 (2.6) | 0.535 |
| $Al_2O_3$ | 13.0 (0.2) | 13.2 (0.2) | 13.1 (0.3) | 13.2 (3.4) | 13.1 (0.4) | 0.987 |
| $Fe_2O_3$ | 3.8 [bc] (0.3) | 3.5 [c] (0.3) | 5.6 [ab] (0.8) | 3.5 [c] (0.2) | 5.8 [a] (0.2) | <0.001 |
| MgO | 0.9 (<0.1) | 0.9 (<0.1) | 1.1 (0.2) | 1.1 (0.1) | 1.1 (0.1) | 0.113 |
| CaO | 2.4 (0.1) | 2.2 (0.1) | 2.4 (0.1) | 2.5 (0.1) | 2.3 (0.1) | 0.236 |
| $Cr_2O_3$ | 0.1 (<0.1) | 0.1 (<0.1) | 0.1 (<0.1) | 0.1 (<0.1) | 0.1 (<0.1) | 0.355 |
| $K_2O$ | 2.6 (0.1) | 2.5 (<0.1) | 2.6 (0.2) | 2.5 (0.2) | 2.4 (0.1) | 0.372 |
| $Na_2O$ | 3.0 (0.1) | 2.9 (0.1) | 2.8 (0.1) | 2.8 (0.2) | 2.6 (0.1) | 0.115 |
| MnO | 0.5 [a] (<0.1) | 0.6 [a] (<0.1) | 0.7 [a] (<0.1) | 0.6 [a] (<0.1) | 0.9 [b] (0.1) | <0.001 |
| $TiO_2$ | 0.1 (<0.1) | 0.1 (<0.1) | 0.2 (0.1) | 0.1 (<0.1) | 0.1 (0.1) | 0.052 |
| $P_2O_5$ | 0.2 (<0.1) | 0.1 (<0.1) | 0.2 (0.1) | 0.2 (0.1) | 0.1 (<0.1) | 0.146 |
| $V_2O_5$ | 0.1 (<0.1) | 0.1 (<0.1) | 0.1 (<0.1) | 0.1 (<0.1) | 0.1 (<0.1) | 0.061 |

### 3.2.2. Carbon and Macronutrients

Total soil pools for C and the macronutrients followed the order of C > N > Ca > K > P > Mg, except in fir stands where P > K (Figures 4 and 5), and this generally matched the order of litterfall inputs of these elements (Figures 1 and 2). Elemental pools in the organic and

mineral soils combined did not differ among stands for these elements (Figures 4 and 5). When analyzed separately, differences among stands were notable in organic (L and FH) horizons, but less pronounced in mineral soils (Figures 4 and 5, statistical test results shown in Table S3). Within the organic horizons, soil C pools were generally higher in hemlock and fir stands compared with pine and deciduous stands (Figure 4). Similar trends were observed for the other macronutrients (Figures 4 and 5), despite lower nutrient inputs through litterfall in conifer stands (Figures 1 and 2). Pine stands had a low organic horizon soil mass resulting in smaller elemental pools (Figures 4 and 5), in conjunction with low litterfall elemental inputs (Figures 1 and 2). Soil P pools were larger in fir stands compared with all other stands (Figure 4), particularly in the FH–horizon. This trend was consistent with large P litterfall inputs in fir stands (primarily through woody debris; Figure 1) although fir stands also had the highest P concentrations among all stands for each litter type (Table S6).

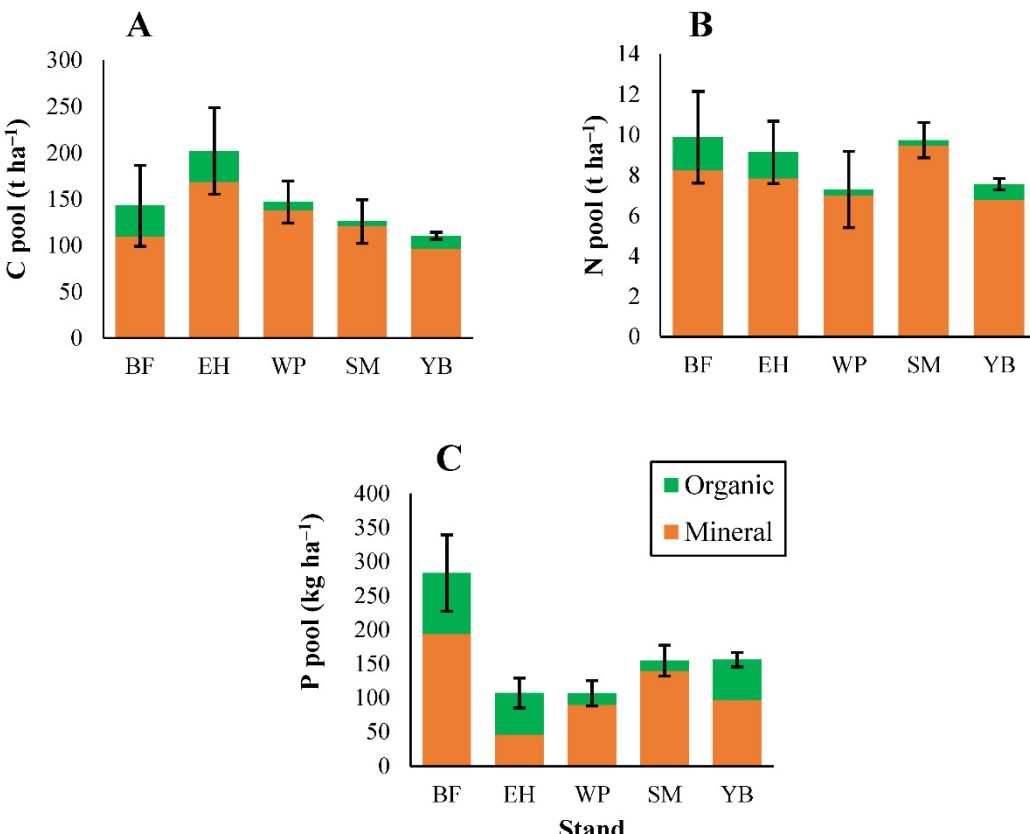

**Figure 4.** Organic and mineral horizon soil pools of carbon (**A**), nitrogen (**B**), and phosphorus (**C**). ANOVA test results shown in Table S3 (no significant differences in the combined organic/mineral pools were present among stands for these elements). Error bars indicate S. E. For BF, EH, and YB, $n = 3$. For SM and WP, $n = 5$.

### 3.2.3. Micronutrients and Trace Metals

Organic horizon pools of micronutrients and trace metals mainly followed the order of litterfall inputs, with some differences (e.g., Al > Mn in some stands; Figure 6). Similar to the macronutrients, larger litterfall inputs did not always translate into greater elemental pools in the organic horizons. Forest floor Al pools were high in birch stands (Figure 6) despite low inputs of Al in litterfall compared with conifer stands (Figure 3). These large Al pools in birch stands were linked to elevated Al concentrations in organic horizons (Table S11) rather than soil mass (Table 2). For Al, 2–17% of the organic pool was in exchangeable form, while for Mn, 30–97% was exchangeable (Figure 6). Differences in soil concentrations were less important for organic Mn pools, which were greatest in fir stands (Figure 6) where organic horizon soil mass was high (Table 2). Zinc and Cd pools were

elevated in birch stands (Figure 6), which also had greater litterfall inputs of these metals compared with the other stand types (Figure 3). Pine stands often contained the smallest trace metal pools, particularly for Sr and Cu (Figure 6), and pine stands were typically associated with the smallest litterfall inputs of trace metals, except for Al (Figure 3).

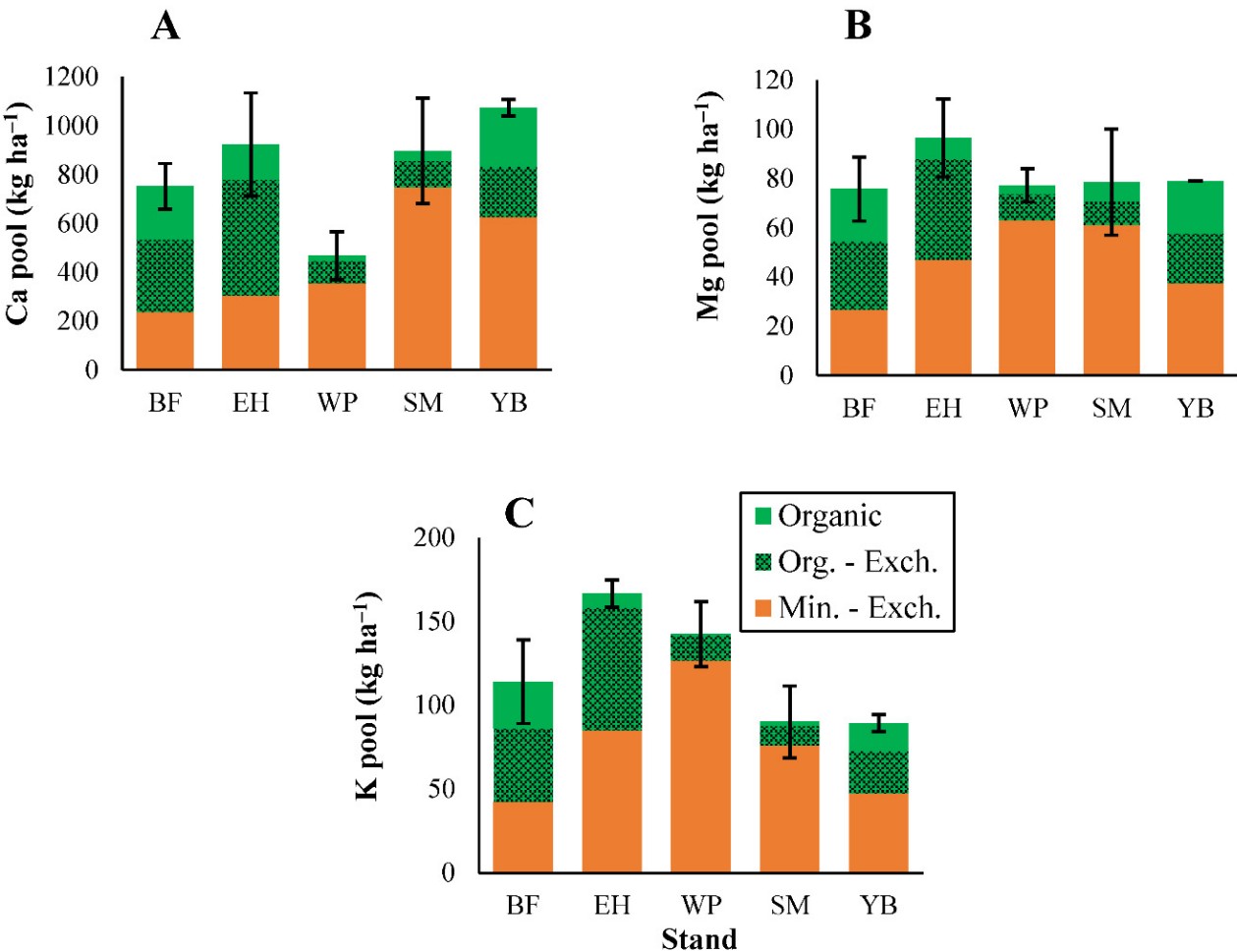

**Figure 5.** Organic (total and exchangeable) and mineral (exchangeable) horizon soil pools of calcium (**A**), magnesium (**B**), and potassium (**C**). ANOVA test results shown in Table S3 (no significant differences in the combined organic/mineral pools were present among stands for these elements). Error bars indicate S. E. For BF, EH, and YB, $n = 3$. For SM and WP, $n = 5$.

### 3.2.4. Organic Horizon Elemental Residence Times

Litter mass residence times in the organic horizons averaged 32 years, but median residence times for all elements were lower in deciduous stands compared with coniferous stands (Table 4). This trend was observed in conjunction with lower organic horizon C/N ratios in maple and birch stands. Maple stands had the shortest median elemental residence times (8.6 years), and hemlock had the longest (76 years). Organic horizon elemental pools were not consistently linked to litter inputs for all elements ($R^2 < 0.21$; data not shown) except for P ($R^2 = 0.57$; data not shown). There were also notable differences in residence times among elements (Table 4). Base cations had the shortest residence times ($\leq 20$ years) while trace elements had much longer residence times, and Al had the longest residence time among all elements (median = 540 years). Both Zn and Cd were cycled rapidly through the organic horizons in deciduous stands, and for Cd, residence times were an order of magnitude greater in hemlock and pine stands compared with maple and birch.

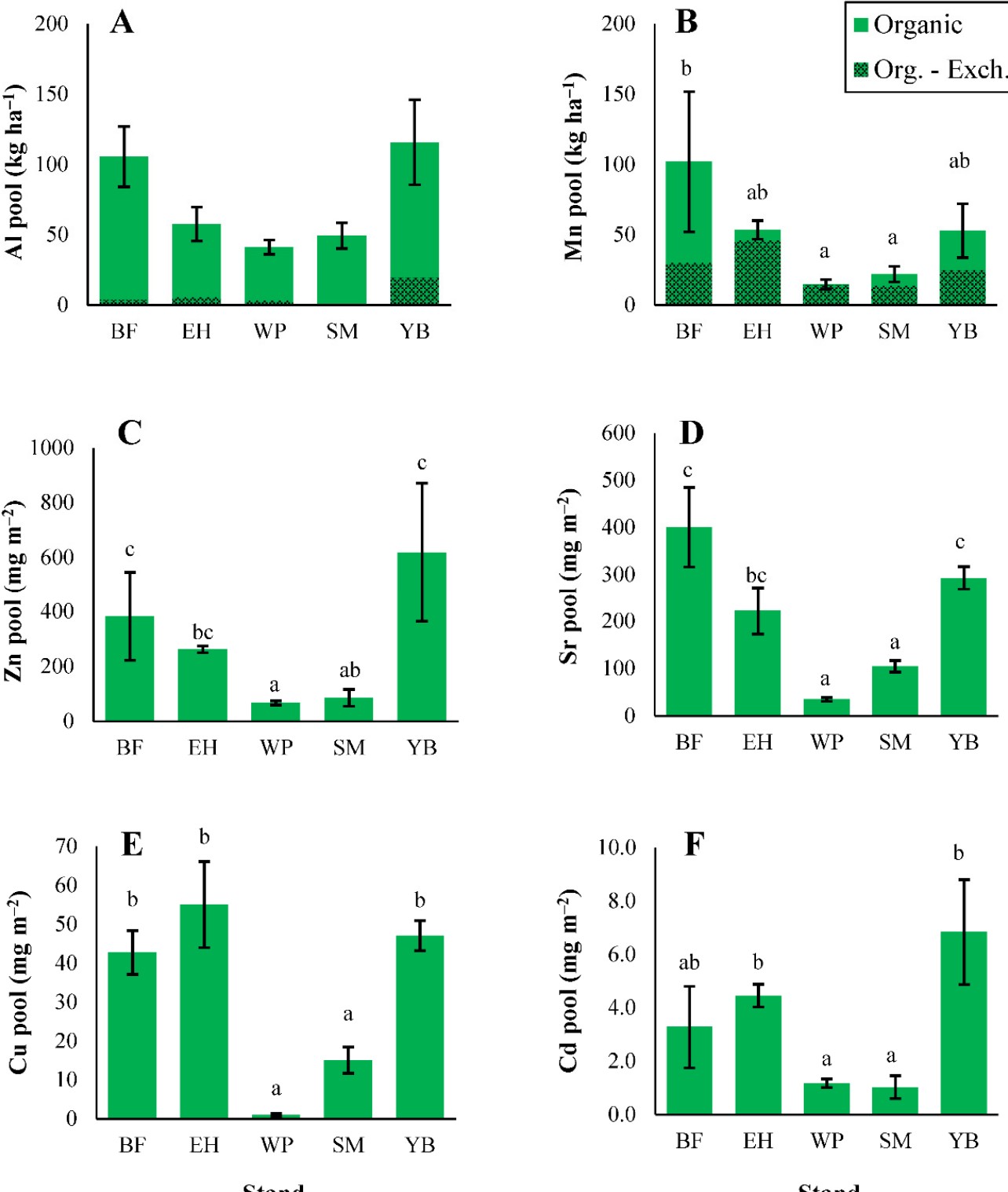

**Figure 6.** Organic horizon soil pools of aluminum (**A**), manganese (**B**), zinc (**C**), strontium (**D**), copper (**E**), and cadmium (**F**). Different letters indicate significant differences ($p < 0.05$) in organic elemental pools among stands; where letters are not present, significant differences do not exist. ANOVA test results shown in Table S10. Exchangeable pools are also shown for aluminum and manganese (darker green). Error bars indicate S. E. For BF, EH, and YB, $n = 3$. For SM and WP, $n = 5$.

**Table 4.** Organic horizon C/N and residence times within the five stand types. Different letters represent significant differences among stands. For BF, EH, and YB, *n* = 3. For SM and WP, *n* = 5.

| C/N or Element | Balsam Fir | Eastern Hemlock | White Pine | Sugar Maple | Yellow Birch | *p* Value | Mean | Median |
|---|---|---|---|---|---|---|---|---|
| | | | Units = years | | | | | |
| C/N–L | 25 ᵃ | 37 ᵃᵇ | 40 ᵇ | 26 ᵃ | 23 ᵃ | <0.05 | 30 | 26 |
| C/N–FH | 20 ᵃ | 26 ᵇ | 26 ᵇ | 19 ᵃ | 18 ᵃ | <0.001 | 22 | 20 |
| Mass | 32 ᵇ | 90 ᶜ | 19 ᵃ | 6.2 ᵃ | 12 ᵃᵇ | <0.001 | 32 | 19 |
| C | 26 ᵇ | 82 ᶜ | 15 ᵇ | 3.6 ᵃ | 10 ᵃᵇ | <0.001 | 27 | 15 |
| N | 55 ᵇ | 105 ᶜ | 43 ᵇ | 9.9 ᵃ | 16 ᵃᵇ | <0.001 | 51 | 55 |
| Ca | 20 ᵃᵇ | 44 ᵇ | 17 ᵃᵇ | 4.1 ᵃ | 13 ᵃᵇ | <0.05 | 20 | 17 |
| Mg | 18 ᵃᵇ | 26 ᵃᵇ | 34 ᵇ | 4.3 ᵃ | 6.9 ᵃ | <0.01 | 18 | 18 |
| K | 15 ᵃᵇ | 46 ᵇ | 22 ᵃᵇ | 2.0 ᵃ | 12 ᵃᵇ | <0.05 | 19 | 15 |
| P | 35 ᵃᵇ | 67 ᵇ | 12 ᵃ | 44 ᵃᵇ | 31 ᵃᵇ | <0.05 | 38 | 35 |
| Al | 624 ᵃᵇ | 182 ᵃ | 173 ᵃ | 540 ᵃᵇ | 1340 ᵇ | <0.01 | 572 | 540 |
| Mn | 34 ᵃᵇ | 78 ᵇ | 17 ᵃ | 11 ᵃ | 15 ᵃ | <0.05 | 31 | 17 |
| Zn | 34 ᵃᵇ | 80 ᵇ | 24 ᵃ | 10 ᵃ | 13 ᵃ | <0.01 | 32 | 24 |
| Sr | 29 ᵃᵇ | 42 ᵇ | 30 ᵃᵇ | 4.6 ᵃ | 18 ᵃᵇ | <0.05 | 25 | 29 |
| Cu | 40 ᵃᵇ | 107 ᵇ | 17 ᵃᵇ | 7.6 ᵃ | 32 ᵃᵇ | <0.05 | 41 | 32 |
| Cd | 62 ᵃᵇ | 158 ᵃᵇ | 242 ᵇ | 16 ᵃ | 19 ᵃ | <0.05 | 99 | 62 |
| Mean | 83 | 82 | 58 | 55 | 127 | | | |
| Median | 34 | 76 | 23 | 8.6 | 17 | | | |

## 4. Discussion

This research showed that litterfall elemental inputs and soil chemistry differed considerably among five common tree species in Central Ontario, Canada. Differences in litterfall elemental inputs were primarily linked to litterfall mass. Elemental inputs were typically greatest in deciduous-dominated stands (maple and birch) and lowest in pine. In birch stands, elevated concentrations of N, Mg, Zn, and Cd also strongly governed litterfall inputs of these elements, in addition to litterfall mass. Soil chemistry and elemental pools also differed among stands but for the most part, soil elemental pools were not directly linked to litterfall inputs. Organic horizon elemental residence times were often lowest in deciduous-dominated stands and for the most part, macronutrients (particularly base cations) had lower residence times than trace metals. As forest composition changes favoring the dominance of deciduous species, elemental cycling is expected to be more rapid for many nutrients (particularly base cations) and trace metals.

### 4.1. Litterfall Chemistry

Litterfall mass and the distribution of litter types varied substantially among stands, strongly influencing macronutrient (and trace metal) inputs in litterfall. Pine stands had the lowest total litterfall inputs and maple and birch had much greater inputs, while fir stands had the greatest inputs of woody debris. Litterfall mass in maple stands was similar to that of an old growth (>120 years old) sugar maple forest in Central Ontario, where litter was collected year–round (3730 ± 294 kg ha⁻¹ year⁻¹) [12]. As well, the average litterfall mass in pine stands in Haliburton was similar to that of a Southern Ontario 65–year–old plantation [34], suggesting that the short collection period captured the majority of annual litter inputs. While the litter collection period did not encompass the entire year, it represents the time when the majority of annual litterfall inputs reach the forest floor [12,31], but these inputs may slightly underestimate annual values. In addition, while these results provide comparisons of litterfall inputs among stand types, year-to-year inputs may vary due to weather events or other factors [35]. In fir stands, a large proportion of the litterfall was from deciduous leaf litter and woody debris (58%). These stands may best represent the present day mixed–species forest composition in this region. Large inputs of fine branches from balsam fir have also been noted elsewhere [36].

Litterfall C and macronutrient inputs were greatest in deciduous-dominated stands and lowest in pine, similar to patterns in litterfall mass. Carbon concentrations were greatest in litter types from coniferous-dominated stands and lowest in maple stands, but because of the much larger litterfall mass, litterfall C inputs were significantly greater in maple stands. Similarly, Neumann et al. [37] found that although C concentrations were higher in conifers across European forests, C inputs were greatest in broadleaf forests where litterfall mass was greater. For the nutrients that are resorbed prior to leaf senescence (N, P, and K) [15], concentrations were typically greater in woody debris than in coniferous or deciduous litter within stands. This was particularly evident in fir stands, where the contribution of woody debris to litterfall inputs of N, P, and K was high. Balsam fir has relatively high nutrient demands (N, P, K, Ca, Mg) compared with other conifers [36] and the greater proportion of woody debris collected in fir stands may account for these large macronutrient inputs. In contrast to N, P, and K, both Ca and Mg are not resorbed to a great extent into woody biomass prior to leaf senescence [32,38]. For these nutrients, concentrations in woody debris were typically similar, or lower than those in coniferous and deciduous litter, contributing to smaller inputs in woody debris.

While litter mass was an important factor that influenced elemental inputs among stands, differences in elemental concentrations were occasionally significant among stands, which also influenced litterfall elemental inputs. Among the five stands, Ca and Mg concentrations were significantly lower in litterfall in pine stands, which along with low litterfall mass led to significantly lower base cation inputs. Pine stands typically have lower nutrient concentrations in biomass compared with other temperate forest tree species [36,39,40]. In birch stands, higher concentrations of Mg led to greater litterfall Mg inputs compared with maple [40,41], despite similar litterfall mass in these stands. Munro and Courchesne [42] found that concentrations of Ca were similar between fresh yellow birch and sugar maple foliage, while Mg concentrations were significantly greater in yellow birch foliage than sugar maple. Similarly, Morrison [41] found that Mg concentrations in foliage were approximately two times greater in yellow birch than in sugar maple. Like Mg, birch stands also had higher N concentrations in deciduous leaf litter and woody debris compared with the other stands [31,40,43].

Litterfall is an important mechanism for the return of base cations to the forest floor. In this study, annual base cation inputs in litterfall were often greater than inputs from mineral weathering or atmospheric deposition. This was especially evident in maple stands, where litterfall inputs were 34 kg Ca ha$^{-1}$, 3.6 kg Mg ha$^{-1}$, and 7.0 kg K ha$^{-1}$ over the collection period. In comparison, average base cation deposition rates at the nearby Plastic Lake catchment from 2007–2017 were 2.9 kg Ca ha$^{-1}$ year$^{-1}$, 0.5 kg Mg ha$^{-1}$ year$^{-1}$, and 0.9 kg K ha$^{-1}$ year$^{-1}$ (Dorset Environmental Science Centre data, Ministry of Environment, Conservation, and Parks), while average weathering rates previously estimated for soils in the Haliburton Forest were 9.1 kg Ca ha$^{-1}$ year$^{-1}$, 3.1 kg Mg ha$^{-1}$ year$^{-1}$, and 3.4 kg K ha$^{-1}$ year$^{-1}$ [44]. This indicates the importance of biological cycling of base cations in replenishing soil pools.

Trace metal patterns in litterfall were similar to those exhibited by the macronutrients, with differences among stands influenced primarily by litterfall mass, and by elemental concentrations for some metals. Aluminum inputs in litterfall differed significantly among stands, while Mn inputs did not. The greatest litterfall Al inputs were in hemlock stands, followed by pine and fir. Similarly, at nearby Plastic Lake, hemlock had the greatest concentrations of Al in foliage among the tree species analyzed, followed by pine and fir [13]. Both pine and hemlock are known to create acidic environments (pH < 4.5) which mobilizes Al increasing its bioavailability [17,18,20]. For both Zn and Cd, concentrations in leaf litter and woody debris were elevated in birch stands. Trace metals also vary in the extent to which they are resorbed into perennial tissue prior to leaf fall. For example, Cu and Mn are well conserved by sugar maple and Zn is more poorly conserved [12]. For Zn and Cd, litter inputs were significantly elevated in birch stands due the combination of high litterfall mass and elevated concentrations of these metals. Both yellow birch and

white birch (*Betula papyrifera* Marsh.) are known metal accumulators, and often have greater concentrations of Zn and Cd in foliage [45–47]. Gosz et al. [15] found that an increasing proportion of yellow birch in forested stands led to greater Zn inputs in litterfall, as Zn is immobile and accumulates in leaf litter over the growing season.

Litterfall inputs of trace metals were typically greater in deciduous stands compared with coniferous stands. Inputs in pine and hemlock stands were similar to those previously determined at the nearby Plastic Lake catchment, which is mainly composed of hemlock, white pine, and red maple (*Acer rubrum* L.) [13]. Trace metal inputs in litterfall were typically greater than bulk deposition inputs [13] indicating the overall importance of litterfall as a mechanism for trace metal cycling in forests, which has also been suggested by others [12,13,48].

*4.2. Soil Chemistry*

The presence of very few significant differences in mineral soil (lower Bm and C horizons) elemental oxide concentrations among stands ($p > 0.05$ for all oxides except $MnO$ and $Fe_2O_3$) suggests that these stands are established on similar parent material. While subtle differences in soil elemental oxide composition have been found to influence canopy tree species distribution [21], differences in soil properties between conifer and northern hardwood stands have also been noted even with establishment on similar parent material [49]. It has been suggested that species-specific litterfall properties can have a strong influence on soil chemistry [17]. Soil forming factors including climate do not vary considerably across this study area due to the close proximity among plots (<30 km), while relief was also similar among stands (slope < 5°). Additionally, stands had been free from recent harvesting disturbances, and regeneration of the species of interest was often observed in the understory, suggesting that these stands have had similar overstory composition for some time. These factors suggest that differences in soil chemistry are strongly influenced by litterfall chemistry, which differs depending on forest cover.

Soils in coniferous stands were more acidic than deciduous stands, throughout the soil profile. In addition, conifer-dominated stands generally contained more organic matter than deciduous stands. The release of organic acids through organic matter decomposition reduces soil pH, and as pH declines below 4.5 [50,51], more Al is mobilized in mineral soil, competing with base cations for binding sites [52]. Other studies have noted substantial differences in soil acidity among tree species. For example, soils beneath sugar maple are generally less acidic than soils beneath hemlock [18,53]. In contrast, it has been suggested that differences may be more subtle, or that changes in forest composition may influence soil chemistry at a slower rate than expected. In Arkansas, changes in soil chemistry were investigated following a large increase in conifers over a 50-year period [54]. Despite greater nutrient concentrations in deciduous (*Quercus* spp., *Acer* spp., and *Ulmus* spp.) forest floor litter and increased acidity of coniferous (*Pinus echinata* Mill. and *Pinus taeda* L.) forest floor needles, there was no difference in mineral soil acidity between these forested stands.

While it was predicted that soil chemistry would differ throughout the soil profile among stands, elemental pools differed most notably in the organic horizons for all elements. Hemlock stands often had large elemental pools in organic horizons due to the large soil mass. Finzi et al. [18] noted negligible differences of Ca and Mg in the forest floor among six tree species but found that Ca and Mg were significantly greater in the upper 7.5 cm of mineral soil beneath sugar maple than eastern hemlock. They also found that Al pools in the same upper mineral soil were almost three times greater beneath hemlock than maple. Richardson and Friedland [22] noted that coniferous-dominated stands often had smaller organic elemental pools than deciduous stands, while mineral soil elemental pools were similar between stands. In this study, organic elemental pools were only occasionally greater in deciduous compared with coniferous-dominated stands. For example, Zn and Cd pools were greatest in birch stands, significantly in the FH horizon. Munro and Courchesne [42] noted the ability of yellow birch to influence soil chemistry, particularly the content of Zn and Cd in organic horizons. In areas of high yellow birch

density, they found that the F–horizon was enriched in Zn and Cd, while the upper B horizon had low concentrations, suggesting that Zn and Cd are taken up preferentially from mineral horizons and returned through litterfall to the soil.

Soil chemistry differed significantly among stand types but not in a consistent way that was observed in litterfall inputs. Elemental pools in organic horizons were associated with litter quality (C/N) rather than the magnitude of litterfall inputs. Macronutrient and trace metal pools were typically smaller in organic horizons in deciduous-dominated stands along with lower C/N compared with coniferous-dominated stands. Litter C/N influences the rate of decomposition and the release of elements to deeper soil [55–57] and N cycling is influenced by tree species [56]. The lower C/N ratio in deciduous-dominated stands suggests that adequate N is available for the microbial decomposition of organic material, allowing mass loss and mineralization to proceed more rapidly than in coniferous-dominated stands [11]. In Wisconsin, the most resource rich sites in an upland forest ecosystem (which included sugar maple) had greater litterfall N inputs, higher litter quality (lower C/N), and shorter N and organic matter residence times compared with resource limited sites [58].

Organic horizon residence times varied greatly among elements and were low for base cations and greater for trace metals. Residence times considered the total organic elemental pool, although for base cations (Ca, Mg, K), most of these pools were exchangeable. Among the macronutrients, base cations had the shortest residence times on average, while N and P had the longest residence times. These patterns are similar to those reported by Gosz et al. [31] who found that base cations had much shorter residence times (range = 2.0–4.8 years) than micronutrients (e.g., Mn, Zn, Cu, range = 5.1–22.5 years) in a maple–beech–birch hardwood forest. In their study, K had the shortest residence time while N and P had longer residence times. Since N and P are translocated into perennial tissues prior to leaf fall, annual litterfall inputs are small compared with organic pools, while K is leached very quickly from litterfall as it is not a structural component of leaf tissue [43]. A large quantity of Ca and Mg returns to the forest floor in litterfall [15], and the need for trees to replenish these macronutrients may lead to their more rapid cycling. In an old growth sugar maple stand in Central Ontario, P also had the greatest residence time and K the shortest [12]. Previous research at the Haliburton Forest suggests that sugar maple growth is limited by available P [59]. The results of this study indicate that P is not cycled as rapidly as the base cations, suggesting that base cations may be limiting tree growth in the region.

Elemental residence times also varied among stands suggesting that species cycle elements at notably different rates. For example, coniferous-dominated stands (especially hemlock) had longer elemental residence times in the forest floor, likely due to slower litter decomposition associated with a higher C/N ratio. In a northern hardwood-hemlock forest in Michigan, hemlocks delivered the fewest base cations in litterfall to the forest floor, but hemlock forest floor pools were significantly greater than for sugar maple or basswood (*Tilia americana* L.), leading to elemental accumulation [60]. Maclean and Wein [61] compared macronutrient residence times between jack pine (*Pinus banksiana* Lamb.) and mixed hardwood stands and found that pine stands had far greater macronutrient residence times than mixed hardwood stands. Richardson and Friedland [22] found that Ca, Mg, K, Mn, Cd, and Cu residence times were greater in coniferous stands than deciduous stands, with K being cycled most rapidly. They concluded that a shift in species composition could substantially alter the distribution of elements in soils over several decades. Finally, residence times of Zn and Cd were short (<20 years) in birch stands, indicating more rapid cycling of these elements by yellow birch [42]. As forest cover changes from higher conifer abundance toward maple dominance, biological cycling is expected to be more rapid for many macronutrients and trace metals.

Mineral soil chemistry occasionally differed among stands but significant differences in elemental pools were not observed as consistently as for the organic horizons. The lack of significant differences for mineral horizons may be due to varying abilities of trees to

obtain elements from lower soil horizons. For example, sugar maple can access nutrients from lower soil horizons through deep rooting systems [53]. Thus, mineral soil elemental pools in maple stands may not be as great as expected with the large litterfall inputs, if maples can successfully take up these elements from deeper soil.

## 5. Conclusions

This study suggests that a shift from a greater conifer abundance to mixed hardwood dominated forests may lead to more rapid nutrient cycling, particularly for base cations. Litterfall elemental inputs varied among stands and were most strongly controlled by litterfall mass (greatest in deciduous sites and lowest in conifer sites). Occasionally, differences in elemental concentrations were also important for explaining differences in litterfall elemental inputs. Similarities in the elemental oxide composition of lower mineral soil (lower Bm and C horizons) among stands suggests that all five stand types were established on similar parent material. Organic horizon elemental residence times were smallest in deciduous stands with a higher pH and lower C/N ratio compared with conifer-dominated sites. Base cations (Ca, Mg, K) were cycled most rapidly, suggesting that they are in high demand by trees and that litterfall is an important component of the biogeochemical cycling of these elements. As forest composition changes due to naturally or anthropogenically induced disturbance events, species replacement can alter soil chemistry and elemental cycling over a relatively short timeframe.

**Supplementary Materials:** The following are available online at https://www.mdpi.com/article/10.3390/f12050613/s1, **Table S1:** Locations and characteristics of the study plots in the Haliburton Forest, **Table S2:** Leaf litter inputs in coniferous, deciduous, and woody debris in stands of five species, **Table S3:** ANOVA test *p* values for litterfall inputs and soil pools for carbon and the macronutrients by litter type, soil horizon, and the sum of total litterfall and the total soil pool. (–) = ANOVA test not conducted for soil horizon, **Table S4:** Carbon concentrations in litterfall and soil beneath the five tree species, **Table S5:** Nitrogen concentrations in litterfall and soil beneath the five tree species, **Table S6:** Phosphorus concentrations in litterfall and soil beneath the five tree species, **Table S7:** Calcium concentrations in litterfall and soil beneath the five tree species, **Table S8:** Magnesium concentrations in litterfall and soil beneath the five tree species, **Table S9:** Potassium concentrations in litterfall and soil beneath the five tree species, **Table S10:** ANOVA test *p* values for litterfall inputs and soil pools for micronutrients and trace metals by litter type, soil horizon, and the sum of total litterfall and the total soil pool, **Table S11:** Aluminum concentrations in litterfall and soil beneath the five tree species, **Table S12:** Manganese concentrations in litterfall and soil beneath the five tree species, **Table S13:** Zinc concentrations in litterfall and soil beneath the five tree species, **Table S14:** Strontium concentrations in litterfall and soil beneath the five tree species, **Table S15:** Copper concentrations in litterfall and soil beneath the five tree species, **Table S16:** Cadmium concentrations in litterfall and soil beneath the five tree species.

**Author Contributions:** Conceptualization: N.F.J.O. and S.A.W.; methodology: N.F.J.O.; formal analysis: N.F.J.O.; investigation: N.F.J.O.; resources: S.A.W.; writing—original draft preparation: N.F.J.O.; writing—review and editing: S.A.W.; visualization: N.F.J.O.; supervision: S.A.W.; funding acquisition: S.A.W. All authors have read and agreed to the published version of the manuscript.

**Funding:** This research was funded by NATURAL RESOURCES CANADA, grant number 311790-2011, to Shaun Watmough.

**Acknowledgments:** We would like to thank the Haliburton Forest and Wild Life Reserve for permitting the research to take place on their property. We thank Adam Gorgolewski, research coordinator, for his assistance in identifying potential suitable locations in the forest for plot establishment, and all who assisted with field and laboratory work. We also thank the reviewers who provided valuable comments on this manuscript.

**Conflicts of Interest:** The authors declare no conflict of interest. The funders had no role in the design of the study; in the collection, analyses, or interpretation of data; in the writing of the manuscript; or in the decision to publish the results.

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
