# Peer review of "Contrasting Litter Nutrient and Metal Inputs and Soil Chemistry among Five Common Eastern North American Tree Species"

_forests, doi:10.3390/f12050613_

Round 1
Reviewer 1 Report
The Manuscript “Contrasting Litter Nutrient and Metal Inputs and Soil Chemistry among Five Common Eastern North American Tree Species” deals with relevant and very important topic of nutrient cycling in forest ecosystems. It is a very well written report based of well conceptualized idea, well executed field work, extensive laboratory analysis, and appropriate data analysis. The presentation of the methods and results is well balanced. In the Discussion the Authors adequately link their finding with the existing work of others. They also point some very interesting and novel findings (e.g. L542-544 the possibility that, in Haliburton forest, the limiting factors might be base cations and not phosphorus, as suggested in previous studies).
I have no major concerns with the paper. There are just some one relatively minor details which Authors might want to consider and few technical issues or typos (see below).
Issue 1:
The Authors state that the litterfall was collected during September-November (L113-114). In the text I didn’t find any reflection on the estimate how much of the litter might have been lost due to the fact that some of the litter was shed during the vegetation season and possible effects on the presented results. Also, the year-to-year variability in litter production might vary (depending on the e.g. wind events. which might increase the shedding of fine branches, but even more so may promote shedding of needles – particularly important if the collection time is relatively short as was the case here). Finally, there is a matter of herbivory and the return of the nutrients to the soil via excrements of animals. The Authors might want to consider reflecting on these issues in the Discussion.
Minor issues.
On several places in the manuscript the exponent or the subscript in chemical formulas or units was written as common text, e.g. L144, L179.
L178: “areal flux” – although it is clear what the Authors mean, I think it is not the accurate term. The “flux” is measured in units of something per unit of area in unit of time (in this year the Authors imply 1 year, but do not state it explicitly).
L225: “0.6 C ha-1” => “0.6 tC ha-1”
L249: I have printed the MS in b-w and the shading of the bars for Coniferous and Deciduous almost indistinguishable. Maybe giving a bit darker colour to one or the other would not be a bad idea.
L261: “(Figure 2).,” => “(Figure 2),”
L575-576: “... residence times were greatest in deciduous stands ...”
I think here is a mistake. Shouldn’t it be “smallest in deciduous stands”?
Reviewer 2 Report
The aim of the authors was to compare the inputs, concentrations and pools of elements in litterfall, and soils in conifer and deciduous forests in Ontario, Canada. Authors measured the elemental composition of main litter input (divided into conifer and deciduous category including woody debris) and soils (whole profile) in eight forest sites. Soil sampling provides a limited information for us as only one composite sample was created per plot, but for main comparison, it is maybe sufficient. In my opinion, such comparisons, although not new, are valuable to other readers.
I lack the idea behind the study in the beginning of the abstract. Why it is so important for us to know the chemistry of litter inputs and the soil and how they differ among all the species? Consider putting one-two sentences about the importance of different species composition of forests and their ecological relevance.
In the introduction part, it would be nice to mention briefly what characteristics generally the soils of natural forests (hemlock and pine-dominated) have and what changes, not only in soil chemistry, can be expected when switching forests to mixed-hardwood forests (complete the idea with 51-61). Is it possible to assume whether this will be rather a “positive or negative” change for the whole ecosystem (e.g. soil acidity; elemental composition; water regime; other species that are connected with this transition…)? This would increase the quality of the manuscript.
For the first paragraphs of the discussion, it would be good to mention the most important differences found between conifers and deciduous trees in general in order to link the text with the introduction. So, what happens with elements and their pools, and total pool of organic matter if the conifer forests turn into forests with maple predominance? And then, go on with the detailed discussion. The same should be done for the conclusion part.
There are no major shortcomings in the text. Some minor I found, I wrote below with individual line numbers. However, in the discussion in section 4.1, there is no overall generalisation of the results found. Occasionally, it was difficult to follow the idea of the text, because of the inappropriate way of comparing data with already published results (e.g. deposition and weathering inputs vs. litterfall input) and it would be good to rewrite this section.
I would like to see a figure or short table with main species representation in % for each plot, it would nicely complete the information given in other figures and table 1.
In the introduction part and throughout whole text, authors often repeat the same word and/or words with the same word basis (even in one sentence). Please, consider substitution of some often-used words with appropriate synonyms. Too many “however” occurred in the results part, and too many “also” occurred in the discussion.
9 remove „chemistry“; remove “differentially”
10 remove “;” and consider using square brackets in combination with parenthesis (the same in the whole text e.g. 103-104
23-26 too long, consider to split the sentence
43-44 this sentence is very general and not well understandable
49-50 explain how it differs briefly, or remove this information
51-52 remove „also“; „Tree species vary widely ….leading to differences …among species“ – please reformulate
107-108 what does it mean recently - in years?
117 Does it mean 4 soil samples? And then according to line 133 one composite sample per plot?
124 Put the information for how long were the soils stored.
126 for 2.3 rather “Litter and soil analyses”
127 put temperature and time, the same for line 132, I suppose 60°C…
150 Why Na+ was not included in sample analyses? Yes, its effect for plant nutrition is minor, but it is important for litter decomposition and thus organic matter storage. It should be explained.
151 and 168 the name and company into parenthesis
211-212 indexes
217 maybe not important but rather considerable?
233 put higher than what…suppose conifers
Figure 1, Figure 2 – if it is mass, then it can not be “flux” in the axis caption (there is no time), use mass or input, or put yr-1
Figure 1 put there a graph showing C/N ratio of the bulk litter and, if possible, also N/P would be nice considering its importance as second limiting soil nutrient - and showed big variation within conifers (BF), and also compared to deciduous species; if suitable the found differences should be included in results and discussed shortly with the same information for soils (Figure 3)
Figure 2B include why statistics is missing
- to all figure and table captions include number of sample replication n=…
273-274 soil pH was generally higher for deciduous than for conifers, nothing new, but should be included
Figure 3B incomplete axis caption
Figure 3 – it should be better to show the information separately, one figure with C, N, P and ratios, second figure with base cations
Figure 4A include why statistics is missing
327 not appropriate term “slight”, remove it
3.2.4. it is not the residence time of organic horizon, but of elements in soil organic horizon, change the caption of this part
351-354 put the information directly like: Organic horizon elemental pools were not consistently linked to litter inputs (R2=…), make this also for other text throughout the manuscript
363-366 remove - start with your main findings, not with general description of the aims of the study
369-370 not well understandable sentence
405-417 what is the main information of this paragraph? I lost myself
421-425 this is not a comparison as there are some numbers only for deposition and weathering, the reader could be confused, put there a comparison with litterfall, or at least sum the input of deposition and weathering…and rewrite the paragraph to more understandable message
430-431 compare total N in deposition not only reactive N forms
451-453 low – also abundant, the sentence is not well understandable
457 these – do you mean given by Landre et al.? Where the study was done?It is not clear to reader, why do you compare your results with them?
483-485 instead of earthworms that generally don´t live in acidic conifer forests, consider the low mean temperature and relatively humic conditions of your sites which both influence the decomposition of litter input
569-570 I don´t agree that this is “new” information, a lot of such studies have been already published
